# Health Service Implementation and Antifragile Characteristics in Rural Communities: A Dirt Research Approach

**DOI:** 10.3390/ijerph20146418

**Published:** 2023-07-20

**Authors:** Samuel Petrie, Paul Peters

**Affiliations:** 1Ted Rogers Centre for Heart Research, University Health Network, Toronto, ON M5G 2C4, Canada; 2Department of Health Sciences, Carleton University, Ottawa, ON K1S 5B6, Canada; paul.peters@carleton.ca

**Keywords:** rural health, health services, ethnography, electronic health, place-based health, antifragility

## Abstract

The implementation of health and care services within rural communities requires necessary sensitivity to the unique facets of rural places. Often, rural service implementation is executed with inappropriate frameworks based on assumptions derived from urban centres. To understand the characteristics of rural communities that can facilitate successful program implementation better, ethnographic accounts of rural health and care services were compiled in rural communities within Canada, Australia, and Iceland. Ethnographic accounts are presented in the first and third person, with an accompanying reflexive analysis immediately following these accounts. Antifragility was the guiding concept of interest when investigating rural implementation environments, a concept that posits that a system can gain stability from uncertainty rather than lose integrity. These ethnographic accounts provide evidence of antifragile operators such as optionality, hybrid leadership, starting small, nonlinear evaluation, and avoiding suboptimisation. It is shown that the integration of these antifragile operators allows programs to function better in complex rural systems. Further, the presence of capable individuals with sufficient knowledge in several disciplines and with depth in a single discipline allows for innovative local thinking initiatives.

## 1. Introduction

Most advanced industrialised nations operate under a universal healthcare paradigm, where all citizens who pay taxes are eligible for basic health care, free of charge [1]. In practice, however, this does not mean that all services are equitably available to all individuals in the same manner. Rather, it principally means that physician- and nursing-based care in the community or hospital is covered under a universal healthcare mandate, while other health and care services exist on a fee-for-service basis [1]. In addition, most specialist care is centralised in only the largest hospitals, primarily located in large population centres [2]. These centres provide the highest level of care for patients and contain trauma centres and specialists who see patients referred from wide geographic areas. The concentration of highly specialized care to certain areas means that service cannot possibly be equally spread out. Citizens who live outside of these specific urban settings have reduced access to health care, despite being assured the same care as other citizens and paying the same income tax [2].

This article focuses on program implementation from communities in three countries with universal health mandates: Australia, Canada, and Iceland. While there are macro-level differences amongst these countries, our analysis shows that there are clear similarities in the selected communities that strengthen our comparative approach. Three separate ethnographies were conducted over a two-year period that can help us understand the presence and impact of antifragile operators and institutional investment indices in rural communities. The results of these ethnographies provide evidence for a foundation on which a place-based rural health system can be constructed, addressing the access gap between rural and urban communities.

Rural communities have long been the centre of agriculture and industry [3]. In Canada, for example, the growth of industries such as logging, the fur trade, and mining have helped build the foundation of development that Canada continues to enjoy today. This rural resource extraction has, indeed, driven Canada’s international reputation as a destination of immigrants since colonisation by Europeans began [3]. In contemporary times, there has been a steady trend of urbanization that has left many rural communities struggling to maintain their viability [4]. While differing in specifics, these trends are shared in Australia, Canada, and Iceland, where our ethnographic accounts are drawn from. These demographic, economic, and political shifts produce an environment where most health policy is dictated from an urban seat—including policies ostensibly serving rural communities [5]. 

It is well-known that health outcomes in urban centres can vary from neighbourhood to neighbourhood. The same is true in rural regions, where health outcomes can vary from community to community. There are notable discrepancies in health service delivery and health outcomes between rural and urban communities as well as between rural communities themselves [6]. Central to these differences in health outcomes are the inequities in efficient access to health services exacerbated by geographic distance [7,8,9]. While communities in Canada, Australia, and Iceland vary in their degree of rurality, even communities within driving distance of larger care centres can still see patients struggle to access basic required services [10]. Health policies and program implementation developed in urban centres that do not recognise these fundamental differences only serve to exacerbate inequities in access and outcomes.

Place-based health is a paradigm of health policy suggested by the Canadian Medical Association [CMA] as a priority moving forwards post-COVID-19 [11]. Place-based health recognises the need for localised geographic approaches to health service delivery and policy that address niche challenges that blanket policy fails to account for. Aiming for the temporal goal of 2040, the CMA is seeking to create a health system [11] that “*is sustainable, more accessible and patient partnered*”, adjusting medical culture so it “is focused on physical and mental well-being and one that embraces equity and diversity”, and, lastly, contributing to a society where “every individual has an equal opportunity to be healthy”. In order to achieve these admirable goals, policymakers have to integrate diverse evidence forms—including qualitative ethnographies—to facilitate place-based health policy suitable for rural communities [12].

Place was considered by other researchers and is a central concept in health system access and interaction, especially in rural settings [13]. From an ethical perspective, respecting the place a patient is from benefits both providers and patients and shifts the treatment paradigm from paternalistic to patient-centred [14]. People have connections to a place for a diverse number of reasons, and this connection has an impact on their health. Referred to as place attachment by psychologists, the role of place in crafting effective and nuanced health policy was highlighted by rural health ethicists [15]. From this perspective, place is defined as the geographical environment that people feel a poignant connection to and have meaningful interaction with [16]. This place-based connection is underappreciated when implementing health service interventions, when developing treatment options for patients, or when healthcare professionals provide therapy recommendations.

To justify and support health program implementations, evidence-based policy has long been considered a gold standard [17]. Via this approach, evidence is accumulated, and rules for implementation are dictated and then applied in a linear fashion, transforming a conceptual problem into a procedural one [18]. However, health systems do not operate in a linear manner and are instead complex systems where the rules of operation are constantly changing and are not directly related to the observed outcomes [19]. Instead, health systems are complex and dynamic, where intervention X may not address outcome Y and be measurable by variable Z in a simplistic manner [20]. The concept of antifragility is useful in this context, where programs are designed to work with complexity rather than against it [17].

### Antifragility and Implementation Sciences

Antifragility is a valuable conceptual framework to guide rural implementation teams to better realise a given initiative. This concept posits that when programs or units are antifragile, they stand to gain from inevitable volatility rather than be harmed by it [21]. This concept was applied in other settings outside of program implementation, such as improving health systems in the wake of the COVID-19 pandemic [22] or as a distinguishing characteristic between two healthcare systems in Eastern Europe [23]. Indeed, by introducing antifragility to a project, implementation teams may improve their ability to scale and sustain in other contexts. 

In the implementation of rural health initiatives, five antifragile operators were identified: optionality, nonlinear evaluation, hybrid leadership, starting small, and avoiding suboptimisation [Table 1]. These operators were derived from an extensive literature review and synthesis [publication forthcoming] as well as a case study analysis of three rural health interventions in diverse contexts [24].

The presence of one or more of the antifragile operators is correlated with an increased adaptability to system stresses, so, by integrating antifragile operators into program design, it is more likely that projects thrive in a complex system, such as healthcare systems [24].

Antifragile operators alone do not signify a successful project. There also needs to be considerable institutional investment. In rural implementation, five institutional investment indices were identified: follow-up and support, policy alignment, champions, utilisation, and funding [Table 2]. The presence of one or more of these indices—correlated with antifragile operators—signals a project that not only possess an ability to manage system perturbations but also has the investment to see sustained implementation. These indices were derived from an extensive literature review and synthesis [publication forthcoming].

## 2. Materials and Methods

This article presents the results of 3 ethnographic vignettes based on direct observation, informal conversations, and town halls completed over a 2-year period [before and during the COVID-19 pandemic] as part of a federally funded international research collaboration [see Funding statement]. This research used dirt research methods embedded within an ethnographic research methodology.

### 2.1. Ethnographic Methodology

Ethnographers need to be good craftsmen, to “avoid the fetishism of method and technique” and allow for theory and practice to become part of the craft [25]. Rural communities cannot be placed in a sealed vacuum and fed variables to understand how and why health program implementation reacts to various influences. While ethnography is often invoked as a methodology in qualitative research, identifying certain methods as “ethnographic” is not always appropriate [26]. 

This research utilised a combination of semi-structured interviews, observational protocols, field notes, and town halls as well as informal conversation to inform the effect antifragile operators and institutional investment indices have on rural implementation. The output of deploying these methods [27] are collated in the form of vignettes. The time period spent in communities varies, from a few days to a few months. Vignettes present information in the form of short stories regarding either literal or hypothetical events [28]. 

Through plain [defined as a traditional social science account of events occurring within the investigation of a single case] and enhanced [defined as an account of events occurring within a single case which uses the presentation techniques of a novelist [25]] vignettes, the presence of traits within rural communities that facilitate antifragile implementation, as well as the need for antifragile principles that can help guide implementation, is established.

Ethnography is a particularly powerful method when deployed in rural research, as it can articulate abstract concepts that can influence an implementation [such as attitudes, beliefs, and creativity] [29]. Ethnographic data are a thick description of interactions, phenomena, and sociological relationships that lead to generative cause and effect [such as in implementation]. Dirt research offers an alternative to ethnography, while still maintaining several of ethnography’s core tenants. The central differences between ethnography and dirt research are the emphasis of reflection on informal interactions and conversation and the shorter time period required to complete it.

To understand the impacts abstract concepts such as optionality and avoiding suboptimisation, among others, can have on implementation, ethnographic approaches were seen as insightful methods. Semi-structured interviews and field notes can provide rich descriptions of mechanisms and events. Informal conversation can contextualise implementation mechanisms. 

Articulating the impact both antifragile operators and institutional investment indices can have on rural implementation can help supplement the place-based health system advocated for in this paper. Ethnographic data can, thus, motivate continued funding of projects, identify gaps in service or needs of rural populations, and effectively contextualise quantitative data. Ethnographic data collected as part of a dirt research approach serve as a written account of the cultural life of a social group, organisation, or community, which may focus on a particular aspect of life in that setting.

### 2.2. Dirt Research Methods

The foundational perspective to rural knowledge generation employed to understand implementation was dirt research, which places an emphasis on lived experience of the researcher through ethnographic research residences [30,31]. Rather than being a methodology unto itself, dirt research is a unifying set of methods that can guide rural researchers in research practices, whether qualitative or quantitative [32].

Dirt research is a methodology that first was described by Harold Innis, a pioneering rural geographer in Canada, and has been revived in the Canadian sociological tradition [33]. Innis prioritised deep immersion in the rural research process and established that the assertions one can make regarding relationships or influence can only be so refined without intimate knowledge of the context from which data originate [31]. Dirt research is an attempt to provide “thick description” of phenomenon that might typically be represented only in summary. This process is about appreciating elements beyond the obvious and making associations between different sources of data and different observations [34].

Central to dirt research is its experiential element where resident scholars eat, travel, play, work, and live in the communities in which they conduct their research. Through these entirely human experiences, scholars come to understand rural challenges through a new perspective—not as a researcher analysing census data in an urban university library but as a rural resident subject to the realities of rural living [35]. This process allows researchers to be close to the lived and sensory experiences of rural places—to feel it as deeply as one can [33].

Researchers have an obligation to participants and institutions that knowledge derived from their activities is obtained in an ethical and fair manner. As with all qualitative research methods, researchers need to recognise the risks of breaching confidentiality and trust between research and participant. Under the ethical plan for this project, researchers did not record verbatim quotes or conversations with participants and avoided attributing specific observations or statements to participants in ways that made them potentially identifiable, unless they consented to attribution. The ethical plan was a guiding sense of principles, underpinned by respect and transparency. There is no structure or target in discussions with rural informants, rather the researchers allowed them to organically unfold and retrospectively made sense of them. Participants recognised the positionality of the researchers—we did not obscure or hide our roles as rural researchers. Gathering our data occurred naturally, without parameters on what we were “looking for”. Informants usually approached the researchers with curiosity about our presence, and dialogue continued from there.

Data, in the form of written field notes, quotes, comments, or reflections, were kept secure on password-protected personal devices, or anonymised journals. No audio recorded information was obtained. Data are retained for 5 years, before being reassessed. Analysis of the data occurred through group reflection and recursive–discursive exercises. Ethics clearance for the overall collaborative projects was obtained by the institutional research ethics board prior to the discussions and accounts summarised in the results section.

For the three presented case studies, each used field notes and written reflective summaries of conversations detailed after encounters with locals. In this paper, we make distinctions between viewpoints held by individuals and viewpoints more widely held. Views from participants were then reconstructed as informants’ contributions interpreted by the researchers [35]. While this approach does not centre the direct voices of informants, it attends to the ethical concerns of research in small, rural communities [36].

### 2.3. Vignette Locations

The three vignettes presented here are drawn from communities that have held different political and economic structures throughout their recent history. The names of the communities were changed to protect the rural informants referenced in the vignettes. The first vignette focuses on Thorfjörður, a small village of under 1000 people in northern Iceland. The research team spent 2 days in Thorfjörður, talking with 5–10 rural residents.

The Thorfjörður, Iceland, vignette is based on dirt research, which places emphasis on understanding phenomena through experiential approaches [31]. The unstructured dialogue between researchers and informants and observational data recorded through the vignette [edited to protect privacy] were codified through reflection, recursive–discursive discussion, and journaling. 

Similarly, the Lemur and Ericstown visits discussed in the second vignette employ a dirt research approach to investigation. The research team spent 3 weeks in Lemur and Ericstown and spoke to dozens of rural informants. Like Thorfjörður, the data recorded within the Lemur and Ericstown vignette is the product of unstructured and semi-structured dialogue, observation, and recursive–discursive discussion. The data are then codified through reflection and journaling. The CVTAC vignette is based on observational work completed during the COVID-19 pandemic. Due to logistical problems stemming from the pandemic, the emergence and deployment of the CVTAC was recorded through discussions with the CVTAC team.

## 3. Results

### 3.1. Thorfjörður, Iceland (2019)

Through a brief windshield inspection [wherein the research team obtained a “first impression” of a town through their van windshield], the town seemed to be static if not declining. There were little to no apparent tourist attractions, few people were going about daily chores despite the dry weather, and the dominant industry [fishing] seemed silent. Most homes seemed neat and well-kept, as is the standard in Iceland, with the clear distinction of more affluent-appearing dwellings being situated on a bluff overlooking the town near the harbour below. A lone cafe at the centre of town was open, so we stopped for a coffee. 

Seven North Americans stopping in for a coffee in early October is a bit of a rarity in Thorfjörður, and the owner who operated the cafe was delighted to speak with us and share more about the town. They revealed that the cafe is busier in the summer but also highlighted another season that sees a strong showing of tourists: winter, which attracts surfers. The fjord at Thorfjörður creates perfect conditions for large, rolling waves in the wintertime, and the surfing there is some of the most unique in the world. This was intriguing to our entire group since, on the surface, Thorfjörður appeared to completely shut down in the winter, when fishing fleets resided in the sheltered windbreak of Thorfjörður’s harbour. Little did we realise that one of the largest tourist attractions in northern Iceland could not be seen from our car. 

After we spoke a little longer, the owner elaborated on some of the other things they do in the town. The cafe owner taught Spanish three times a week at the secondary school in Thorfjörður and operated a canteen, where students could purchase meals and refreshments, during the lunch hours. In what we now realised was typical of Icelandic hospitality, they offered to take us to the high school for a tour, an offer that we accepted. 

The fact that an individual has multiple roles in a community [cafe owner, Spanish teacher, and caterer] is not unique to rural Iceland. Individuals in rural communities across the globe exhibit an impressive level of agility. Smaller populations require people to be proficient in a wide array of skills. In an urban centre, these skills are more likely to be siloed into professions and trades. In rural communities, it is often the case that on top of the “official” role people adopt, they have the skills and know-how to operate in many fields. The effect is a level of pliability that is not necessary in larger urban centres. 

Although it is difficult to describe in explicit terms, the flexibility of rural villages allows them to more easily meet challenges than more rigid urban centres. The theme of multiple proficiencies was continued when we met with the school teacher of the secondary school that afternoon. The school teacher, in addition to their formal role, was also a team lead for the volunteer organisation that operates search and rescue missions year round in Iceland.

The school teacher spoke at length about the school, their methodology as its leader, their past, and what they hope Thorfjörður’s future generation can achieve after their tenure at the school. The school teacher had taken the job after the 2008 financial crisis, which heavily impacted Iceland. When they took the role, the school had a low rate of graduation and had seen many past students fail to secure a high school diploma at the end of their schooling. Within a short period, the school was transformed into a model place to study and to work. Almost all students who attend the school now graduate with a high school diploma, and the school has been recognised by the government of Iceland for its excellence. The school teacher had a large influence on the school’s success, encouraging staff to attend international conferences, devising methods to highlight the staff’s strengths, and encouraging teachers to create their own path.

At the conclusion of our meeting, the school teacher told us how they were considering retiring in the coming year. When asked if they were mentoring or teaching one of the staff to fill their role similarly to how they had, their response was fascinating. They asserted that it is not their place to tell anyone, including students, how to approach their work or studies. They are a firm believer that people must determine their own course based on their strengths, interests, and ability. This is a perfect example of embracing the emergent properties that are ubiquitous in human systems. Emergence is evident within healthcare systems as well, and organisations that embrace it and work with it are more successful at implementing interventions than those that do not. 

When we finally shook hands and took a picture to commemorate and close the meeting, the school teacher deftly illustrated how the antifragile aspects of rural communities can allow them to thrive in the face of multiple challenges. The comfort with ambiguity, the incorporation of multiple perspectives in decision making, and accounting for emergence—all contribute to creating an environment primed to capitalise on uncertainty to become stronger. 

We use “rural communities” as a term in broad strokes, as these antifragile properties are common to not only Thorfjörður but also Renfrew County, Canada, and Lemur, Australia. Collaborative efforts can only truly thrive if the people or agents working together can sufficiently address the weaknesses of the other people within their group—something that requires a breadth of knowledge across disciplines. 

### 3.2. Lemur and Ericstown Field Visits (2019)

As we emerged on our first day, we were immediately stopped by the oppressive sun that had only just crested the horizon. It was 8:00 am, and the temperature was already hovering in the low 20s °C, while it was about −20 °C at our university in Ottawa, Canada. We crossed the small creek separating our accommodations from the Lemur town hall, where the field visits were to begin. 

We had been in Lemur for a week already and were pleasantly surprised at the relative bustle within the town early on a weekday morning. Trucks were making deliveries to the several cafes and restaurants on the main road, and people milled about the central square, greeting each other as they passed. The sense of community was strong—even to a casual observer. A pillar that served as the notice board in the middle of the square was crowded with notices. These ranged from farmers looking for locum labourers to the promotion of musical events and reading groups hosted by local groups and establishments. Contrary to the cliched understanding of a rural town as a place that has been passed by, the cultural, social, and economic connections and relationships were clear from just a simple bulletin board. 

Our group had gathered and were to visit Ericstown, which was about 45 km away, for a day-long excursion meant to educate ourselves on the greater context of rural Australia. As we pulled out from in front of the Lemur town hall, an older adult man stood up at the front and introduced himself as the tour guide. He had lived and farmed in Lemur and the surrounding regions his entire life and said he was excited to share its history with us. We took note of the genuine pride and enthusiasm he displayed as well as the sense of ownership he imparted when discussing his experiences within rural Australia. 

Our route followed Goyder’s Line, a geographic border that separates arable land from the beginning of the outback [37]. Land north of Goyder’s Line receives less than 250 mm of rain a year so is generally untenable to farming practices. Geographically, this line represents a contrast between the rural and the Outback, as the townships and villages north of Goyder’s Line are few and far between and rely on the raising of grazing animals [such as sheep] as their main economic driver. South of the line, traditional farming [crops, vegetables, grains, etc.] was more common, and the number of incorporated townships was more prevalent. Along this line there are many abandoned properties, and their ruins still scatter the highway, which largely follows the line from Lemur to Ericstown, a stern reminder of hubris and trying to force the issue of progress against a harsh climate.

As we arrived in Ericstown and disembarked from our bus, we had a chance to stretch our legs, as the roads were in rough shape. The sun was truly at its apex now, and the temperature seemed to be in the low-to-mid 30s. We were to meet a group of local stakeholders who had been influential in several initiatives in Ericstown. Before we ventured into the community hall where the meeting was to take place, our guide offered his opinions on the potential for electronic initiatives to become more common in rural places. He was approachable and handled questions with the straightforward nature he had developed following years “in the bush”. When asked about electronic health projects and his general thoughts, he frowned. 

“Ericstown used to have a bank branch here, you know. Only branch for several hundred kilometres.” He scoffed, “Until, of course, online banking came along. We were told everything we could do at the branch could now be done online. Of course, that’s not true. The branch packed up a week later. Shame too, it was the most handsome building in Ericstown.”

He motioned to an early 20th century brick and stone building in the centre of the town. It was clearly vacant, though still retained the charm that our guide referred to.

The hall was full of people, mostly older and mostly women, who seemed excited to receive us. It was a largely unstructured event, as conversations broke out across the hall after some brief introductions. Our group sought out informal opinions regarding electronic health programs from several incredibly kind older women who had clustered in the middle of the hall around a table. After rebuffing several of their attempts to provide refreshments or various pastries, they shared their thoughts. 

One community member said that electronic access is great—“I used one of their tele services the other week when I visited the hospital.” However, another community member observed that, “I think telehealth is an opportunity to move nurses around. That’s what happened in another community close to here, don’t you know. They introduced a telehealth service, and the nurse was taken away from the community!” For the second time that day, we learned that electronic initiatives do not always correlate with progress. Another woman echoed the sentiment of the second community member that electronic health is fine—“just so long as they don’t use it as an excuse to take services from us and redistribute them.” After a second round of coffee, tea, and croissants that were politely refused, our group said our goodbyes and exchanged information with several of the stakeholders. We boarded the bus and prepared for a return trip to Lemur. 

The return trip offered a chance to debrief and reflect on the conversations of the day and what the peer trainees in our group had learned. Conversations bounced across the bus at an excited pace, as connections and meanings were discussed based on the thoughts and concerns of these rural stakeholders. When we finally pulled back into the front of the Lemur town hall, our group was motivated by the events of the day yet challenged by how our preconceived notions on electronic health benefits were confronted.

### 3.3. CVTAC in Canada (2020)

In the summer of 2020, our research group was forced to adapt our methods of in-person community engagement given the extensive lockdowns in our partner countries of Australia, Canada, Iceland, and Sweden [29]. As such, we began to engage in autoethnographies of our own communities to better understand how rural places were responding to the pandemic. In an all-hands meeting early that summer to outline the methodological approach to these autoethnographies, we agreed that the goal of these ethnographies was to better understand the difference between communities that were actively thriving and those that were desperately coping. The thriving communities were thought to have encouraging practices or initiatives that could then be replicated in other rural places. 

One very encouraging practice that came out of Ontario during the COVID-19 pandemic was the County Virtual Triage Assessment Center [CVTAC], which was developed in an effort to redirect patients from using the emergency department/hospital for things that can be provided by a family practitioner, such as prescription refills. It was hoped that the implementation of the CVTAC would strengthen access to primary care. Its goal was to reduce the demand on the emergency department, and its prolonged implementation can only be beneficial in combatting emergency department overcrowding into the future, post-COVID-19 pandemic. Without the CVTAC, the primary care that was available in rural Ontario was difficult to access before the pandemic and became near impossible during it.

This virtual triaging process was meant to assist rural patients in navigating their symptoms and the system and provide them with the necessary information to understand if a visit with the emergency department was needed. While the implementation of a centre such as the CVTAC is contingent on other factors relevant to the function of a health system [such as high numbers of patients with a family physician who can follow-up with their patients], its initial reception was positive from both patients and providers.

Having sufficient vision, creativity, and resourcefulness to implement a grassroots project such as a virtual triage centre warrants further investigation. COVID-19 required a delineation between rural communities that were desperately coping with rapidly changing conditions and those that were productively flourishing in the face of unprecedented adversity. These abstract facets such as creativity and imagination are central to a community’s innovative capacity. Measuring these concepts is difficult through quantitative approaches, however, and can be more readily understood through qualitative means. Successfully applying local knowledge regarding access challenges served the CVTAC well, and this practice should be disseminated to other rural contexts to aid in generalised eHealth implementation in rural places. 

Evaluating the reception of an eHealth initiative can be difficult, as it can be biased by many factors. One major factor is the reality that there was simply no other service for patients to access that abided by public safety protocols at the time. Emergency departments were kept secure to the point of obstruction, as only the sickest of patients were being admitted with non-COVID-19-related symptoms. Many patients who had relied on the emergency department as a safety net for complex social needs were also seeking care elsewhere. When the implementation of an alternative method of access gains support, the evaluations of such an effort are positive in nature, simply because the result of not receiving access in a timely fashion for emergencies would likely lead to poor outcomes for patients. 

There have been endless calls for moves away from “one size fits all” service models, with assertions that policy that focuses on outcomes [accessibility and health outcomes] is likely to be more effective than policy that focuses on inputs. There also needs to be local leadership and champions, as seen in this implementation of the CVTAC, where imagination and creativity allowed local actors to recognise good ideas when they see them and coordinate their implementation and ongoing operation. 

Local leadership is central to what is known as innovation capacity. Building innovation capacity in rural communities allows for grassroots initiatives and local thinking to develop impactful solutions that suitably address rural nuance and difference. The CVTAC initiative was strengthened by its structure as a project supported by multiple agencies in the community, not just by the hospital where it was housed.

The funding to support the CVTAC was initially tied to COVID-19 and was precarious for some time during the first year of the pandemic. Technology and innovative strategies such as the CVTAC need clear funding sources moving forwards, as creating an inherent clause in their implementation to roll them back post-COVID-19 would be damaging to the overall rural health system they were introduced into. 

Despite uncertain funding, local champions, including the elected provincial representative, successfully advocated for long-term funding for this program. This highlights the importance of having local advocates who can champion initiatives and retain the levels of interest [and funding] necessary for their continuation.

Table 3 summarises the above vignettes and the antifragile operators and institutional investment indices present in the three vignettes. Elucidating the impact of the operators and indices identified in the results occurs in the next section. 

## 4. Discussion

The vignettes discussed in the Results section, the Iceland vignette, Australian vignette, and Canadian CVTAC vignette, provide evidence of the presence of antifragile operators [AOs] as well as institutional investment indices [IIs], as summarised in Table 1 and Table 2 above. Further, the impact of the AOs and IIs can be reasonably inferred from the ethnographic accounts. Implementation within rural communities often struggles to rectify the reductionistic requirements of evidence-based medicine with the realities of rural research [38]. In accordance with suggested ethnographic reporting protocols, the Discussion section of this paper incorporates significant reflexive, recursive, and discursive exercises to tease out the effects AOs and IIs have on rural implementation [27]. 

### 4.1. Antifragile Operators

One of the key aspects of the Thorfjörður, Iceland, vignette is the demonstration by the sociological infrastructure of this rural community to integrate antifragility in a cohesive sense. Having a breadth and depth of knowledge, as well as a comfort in figuring a situation out, as displayed in Thorfjörður, results in a resourceful population capable of innovation. This includes capitalising on the emergence of social organisations that are common in small places, engaging agents that are T-shaped in their proficiencies—meaning they have a breadth of knowledge in multiple disciplines but are especially adept at one, and lastly leveraging the increasing comfort with the ambiguity that rural communities must incorporate into their outlook on implementation efforts.

#### 4.1.1. Optionality

Optionality is simply the ability to make choices. Due to the breadth and depth of knowledge within rural places such as in Thorfjörður, Iceland, they are afforded optionality in response to system stress. Being proficient in several tangible domains—such as languages, management, survivalism, etc.—allows for a requisite diversity in response to paradox and ambiguity. It also allows for suitable understanding and interaction with emergence, a trait common to sociological systems. The school teacher in Thorfjörður displayed their comfort with emergence through the assertion that people cannot be “forced” into one role or another, rather they must be given the tools to make their own choice, and then they naturally gravitate towards a position that best suits their skills, position, and ambition. The comfort with the emergent properties of systems is one that can then be leveraged through optionality in rural health system implementations. 

#### 4.1.2. Hybrid Leadership

Hybrid Leadership is defined as both bottom-up, grassroots influence as well as top-down, traditional guidance on a system’s direction. In the context of an antifragile eHealth project’s design, hybrid leadership requires input from both frontline providers and management-level decision makers.

The collaborative nature of rural places facilitates the integration of multiple perspectives on issues and challenges. In Thorfjörður, Iceland, this signaled the importance of hybrid leadership as well as its presence within the community. By encouraging and enrolling bottom-up knowledge within an implementation, ownership of the intervention can be facilitated, and a greater sensitivity to an intervention’s effects can be accomplished. As Bhandari et al. note: “Several prior studies in community health research involving rural areas have found that positive community health results from professional–layperson partnerships engaged in capacity building, diverse leadership development and comprehensive multi-sectoral planning” [39]. Further, Barret notes: “Clinical drivers and telemedicine users must own their own systems” [40]. Hybrid leadership, as seen through the school teacher’s encouragement of their staff and the multiple roles rural individuals fill within the community, is important for creating accountability within an implementation and community ownership, enabling successful implementations. 

The grassroots nature of the Canadian CVTAC vignette also highlights the importance of mobilising local knowledge through hybrid leadership. The site where the CVTAC was developed is unlikely to be seen as a suitable candidate, due to its rural location and perceived lack of innovative capacity, by urban policymakers when dividing funding for a virtual care initiative. 

Through local knowledge and skills—also seen in the Iceland vignette—as well as capable leadership, the CVTAC arose to address the problems of rural health access at the height of the COVID-19 pandemic. Successfully integrating bottom-up information and drive within an implementation allows for nuanced responses that can successfully address challenges and barriers that are unique to a rural context. 

Employing hybrid leadership modalities is again valuable to the successful implementation of the CVTAC, and measuring the presence of these leadership typologies is best accomplished through qualitative methods. According to Volpe, “thick description involves capturing the meaning and experience in a situation in a rich and detailed manner, creating the conditions for interpretation and understanding” [41]. Qualitative methods are optimised for capturing this thick description. 

#### 4.1.3. Nonlinear Evaluation

Nonlinear evaluation is defined as the surveillance and assessment of an intervention outside of traditional success or failure labels. In the context of antifragile eHealth project design, nonlinear evaluation seeks to understand the impacts an intervention has on the other aspects of the system.

The importance of nonlinear evaluation is seen through the dispersion of complex medical and social patients that occurred when restrictions were imposed on a regional emergency department. As seen in the Canadian CVTAC vignette, patients were taken on by other services that did not have the sufficient expertise nor the resources to effectively treat them, resulting in poorer outcomes. Further, routine visits—such as prescription refills—also became more difficult, risking the progression of otherwise manageable disease. Establishing the CVTAC helped alleviate these problems, which risked cascading throughout the system and threatening its integrity. 

Providing better access to health services for complex patients and routine patients alike through the virtual triage system allowed for evaluating patients across a spectrum of complexity, protecting other services, and maintaining the integrity of the system. Understanding these effects through nonlinear evaluation was an important motivator for the CVTAC, including its presence in this rural health system moving forwards.

#### 4.1.4. Starting Small

Starting small is defined as limiting the initial implementation process to small-scale rollouts. In the context of antifragile eHealth project design, starting small requires an iterative approach, with more components of an intervention included with each iteration of the implementation.

Evidence of the effectiveness of starting small can be seen in the CVTAC vignette. By iteratively implementing the virtual triage intervention in a stepwise manner, the CVTAC team was able to incorporate feedback from multiple streams, avoid committing limited resources to one course of action, and allow for tinkering with the delivery modalities and communication. Coupled with the antifragile operator of optionality, starting small allows implementation teams to better understand how their intervention fairs when confronted with the complexity of in situ environments. Together, they help to guard against path dependence, where projects that have minimal impact on behaviour or outcomes are continuously funded for political or organisational reasons. 

#### 4.1.5. Avoiding Suboptimisation

Avoiding suboptimisation is defined as avoiding the maximisation of efficiency of one aspect of a system at the expense of another. In the context of antifragile eHealth project design, avoiding suboptimisation requires sufficient understanding of the system and influence of the intervention, to account for its changes in system function post-implementation. Implementing electronic health initiatives can have the effect of improving health access to rural communities. As identified, first by our guide and then by the three community stakeholders in the Lemur and Ericstown vignette, this transition to electronic mediums of access can also have unintended consequences. Indeed, instead of being seen as innovative, the provision of service through an electronic option was seen as harmful—first in online banking and again in telehealth, resulting in the redistribution of nursing resources. As Alami et al. note, “the sustainability of [an eHealth initiative] is partly dependent on whether the organizations and other professional actors are able to accept, recognize, and formalize this new role of [the project]” [27]. It is important, then, for implementation teams to monitor the opinions and attitudes of key stakeholders throughout an implementation to avoid suboptimising the entire system. 

This further motivates the integration of hybrid leadership modalities and nonlinear evaluation approaches when deploying antifragility in rural communities. To guard against suboptimisation, appropriate information must be gathered from and interpreted by diverse sources within the system. Indeed, eHealth evaluations that only consider the opinions of senior policy makers or decision makers [who are likely to be heavily influenced by the improved bottom line of the system by replacing a paid nurse with an eHealth system] miss the negative sentiment of the community. This negative sentiment is important to capture, as it was established that attitudes and beliefs are major barriers to the implementation of eHealth initiatives [42]. While a singular eHealth initiative may display “success”, future implementations would likely be unsuccessful, costing the system money [and the patients quality of life] over time.

### 4.2. Institutional Investment Indices 

Institutional investment is the necessary supplement to a program or community’s antifragility. A reality of implementation is that if there is no corresponding investment by relevant institutions, the projects would not realise their full potential. By tempering institutional investment with antifragility, organisations can avoid path dependence, maximise efficiency in their implementation, and ensure future system outcomes are not overly influenced by previous implementations. 

#### 4.2.1. Follow-Up and Support

Follow-up and support is defined as the structures, channels, or protocols prescribed and maintained by institutions and organisations to facilitate support for the eHealth intervention. In the context of institutional investment, it pertains to both patients and providers at the individual [micro] level as well as at the system [meso] level. 

The Thorfjörður, Iceland, vignette provides evidence of how implementation teams can enroll follow-up and support within a community. Usually, outside support is required for an electronic health intervention [such as technological or clinical support]. This vignette shows the versatility of rural populations and the importance of implementation teams performing their due diligence before immediately seeking outside help. Logistical, organisational, and perhaps even technical support and expertise could exist within the community in which the intervention is being implemented. Enrolling this support would further increase the interventional ownership sentiment within the community as well as improve the capacity and repository of skills within the given rural context. This could facilitate easier implementations in the future. As Bradford notes: “The shift away from paternalism and towards self-advocacy and empowerment has occurred across many aspects of society. There has been a societal change with the questioning of authoritative services, a move to more consumerist model with greater accountability and focus on quality” [43]. Enabling rural communities through self-advocacy and shifting from paternalistic support improves the community, or system, as a whole.

Follow-up and support is also displayed as an important institutional investment index for implementation teams to prioritise in the Lemur and Ericstown vignette. When the bank closed its physical branch, an assumption of the banking corporation was that simply because a system is present, it would be used. Instead, consumers begrudged the bank for leaving and found much of the service they were accustomed to was not available in the newer, online medium. Enrolling local support to facilitate the change may have eased the transition, allowing for rural community members to better familiarise themselves with the novel nature of online banking. Ensuring that appropriate follow-up and support structures are present is integral to an implementation’s success, even more so if these structures are local in nature. 

#### 4.2.2. Policy Alignment

Policy alignment is defined as the adherence or reference of an eHealth intervention’s targeted population or outcomes to broader organisational policy. In the context of institutional investment, policy alignment can be seen by the linkage to the initiatives undertaken at other organisations and at the recommendation of provincial or national guidelines.

The CVTAC vignette provides an important example of policy alignment and this institutional investment index within a program. The CVTAC arose in large part because of the COVID-19 mandates in place at the time—including the requirement of almost all services to mitigate face-to-face interaction. Funding for the CVTAC [another institutional investment index] was released dependent on the policy in place at the time. The CVTAC team astutely [and necessarily] aligned their virtual triage initiative with the recommended national policy at the time. This led to increased support and robust funding as well as showcased the CVTAC as an exemplar virtual triage project for other organisations to replicate and follow. 

#### 4.2.3. Champions

Champions are individuals, usually healthcare providers but sometimes patients, who advocate, promote, and support an eHealth initiative’s implementation. In the context of institutional investment, champions are important for influencing institutional investment because of the pressure they can apply through personal and professional relationships. 

In Thorfjörður, Iceland, the school teacher’s advocacy and promotion of electronic means of education led to increased graduation rates and the recognition of their school as one of the best in Iceland. This reaffirms the importance of enrolling a champion within health implementation [or any other implementation, for that matter]. The formal and informal status of a key agent can have beneficial impact on the attitudes and values of other individuals important to a successful implementation. Identifying such an agent should be a priority of implementation teams. 

#### 4.2.4. Utilisation

Utilisation is the tangible rates or quantitative measures of the patient and provider’s interaction with a given eHealth intervention. In the context of institutional investment, utilisation outcomes are important in determining the return of investment that institutions are evaluated by at broader levels of funding. Funding and utilisation are two self-evident indices of institutional investment. Nevertheless, establishing secure funding streams, as well as ensuring that the utilisation of an intervention is effectively captured [employing nonlinear methods if needed, such as supplementing usage rates with semi-structured interviews], is critical to an intervention’s success. The current funding for the CVTAC relies heavily on COVID-19 grants provided by the federal and provincial governments. Maintaining this funding level post-COVID-19 to further derive benefit from the CVTAC should be an important priority of the implementation team. 

#### 4.2.5. Funding

Funding is the monetary commitments by organisations to health investments. In the context of institutional investment, establishing consistent funding sources and schemes is important to determine the confidence an institution has regarding an eHealth project’s ability to impact outcomes. Funding concerns are present in all the presented vignettes. Much of the implementation within rural communities is grant-based. This grant-based nature of rural implementation requires the sufficient prioritisation of sustainability. Ensuring that projects are antifragile, as well as aligned to contemporary policy objectives, improves the probability that a rural project sees sustainment in its community. 

### 4.3. Limitations

To protect the identity of rural informants, their names were changed, and the names of their communities were adjusted. Further, some of the quotes attributed to them were modified. This is necessary, as the small size of rural communities makes anonymisation an important aspect of rural inquiry to avoid potential harm to our collaborators. This also means that the verbatim quotes originally recorded had to be editorialised. This can obfuscate the perception of the informant’s opinion on rural phenomena and implementation. Where possible, the sentiment contained in verbatim quotes was communicated as wholly as possible. Indeed, the approach of using enhanced vignettes captured the information provided by rural research participants but occluded or otherwise adjusted to account for the ethical responsibilities of rural research. As such, there may be sections of information that are subject to the interpretation and bias of the researcher and do not wholly reflect the understanding of the rural context provided by the informants. Dirt research relies on transparency and trust between the researcher and participant. To maintain and promote that trust, the qualitative accounts included in this manuscript had to be adjusted. 

## 5. Conclusions

Representation in rural health research is important and contributes to the growing need for place-based health system reform. Place-based reform could prioritise equitable care where every individual has an equal opportunity to be healthy [44]. Good health is something that is often taken for granted, only missed in its absence. Without good health, work becomes more difficult, relationships become strained, and personal circumstances can deteriorate. To date, rural populations internationally have not had the same opportunity to be healthy as their urban counterparts. Through poorer healthcare access, poorer health outcomes follow [6]. The presence of antifragile operators within rural communities provides implementation teams with a foundation on which to build and, by integrating these operators into implementation processes, better outcomes help afford every rural resident a chance at good health.

The assessment of ethnographic data was a point of debate in the past. Truth is sometimes subjective and can be distorted through research as biases, interpretations, and assumptions are applied to phenomena recorded through ethnographic accounts [25,45]. Ultimately, this paper subscribes to the Humphreys and Watson understanding of ethnographic truth. They posit that while truth exists on a spectrum, truer ethnographic data are accounts that can prepare someone entering the area of life summarised by the account to cope better than by reading a different report [25]. 

The vignettes developed through dirt research and presented here aim to prepare the reader for undertaking an implementation of an eHealth initiative in a rural community. Beginning an implementation of a digital health intervention within a rural community is a difficult undertaking, requiring sensitivity to multiple aspects before effectively communicating these to rural stakeholders. Dirt research can help frame challenges and disseminate successes, particularly when antifragile operators can be integrated into the project.

Our analysis of the presence of and need for antifragile operators and institutional investment indices following these ethnographic accounts helps the reader understand why a principled design philosophy is needed—and why antifragility is an appropriate candidate in that regard. Leveraging ethnographic approaches to understand and appreciate the importance of conceptual facets such as antifragility is crucial to injecting innovative practice into rural health systems, thus strengthening them for future stresses.

Ultimately, this paper shows that rural communities possess important facets that can be leveraged during implementation. These facets include the antifragile operators of antifragility, nonlinear evaluation, avoiding suboptimisation, starting small, and hybrid leadership. Additionally, institutional investment indices such as funding, follow-up and support, policy alignment, champions, and utilisation are important for a program’s longevity. Further, the presence of T-shaped personnel within these communities provides flexible and adaptable support for rural implementation and initiatives. 

Through the tandem employment of antifragile operators and qualified support personnel, rural implementation teams can feel comfortable and confident in engaging with an implementation process. The role of this confidence and comfort is hard to overstate, as the importance of confidence in the approach, confidence in the outcome, and confidence in the ability to recover from mistakes all contribute to better implementation and ultimately better population health outcomes. 

All residents who live within a universal healthcare system deserve an equal chance at health—acknowledging that rural contexts are different from urban contexts and not necessarily disadvantaged is one way to start realising that aim. 

## Figures and Tables

**Table 1 ijerph-20-06418-t001:** Antifragile operators and their definitions.

Operator	Definition
Optionality	The ability to make choices.
Nonlinear evaluation	Qualifying an intervention by rejecting traditional success or failure labels.
Hybrid leadership	Bottom-up, grassroots influence, as well as top-down guidance on implementation direction.
Starting small	Limiting initial implementation process to small-scale rollouts.
Avoiding suboptimisation	Avoiding the maximisation of efficiency of one aspect of a system at the expense of another.

**Table 2 ijerph-20-06418-t002:** Institutional investment indices and their definitions.

Index Item	Definition
Follow-up and support	The structures, channels, or protocols prescribed and maintained by institutions and organisations to facilitate support for the intervention.
Policy alignment	The adherence or reference of an intervention’s targeted population or outcomes to broader organisational policy.
Champions	Individuals, usually healthcare providers but sometimes patients, who advocate, promote, and support an initiative’s implementation.
Utilisation	The tangible rates or quantitative measures of the patient and provider’s interaction with a given intervention.
Funding	The monetary commitments by organisations to a program’s investment.

**Table 3 ijerph-20-06418-t003:** Presence of antifragile operators and institutional investment indices.

Lemur, Australia	Thorfjörður, Iceland	CVTAC, Canada
Antifragile Operators: optionality, nonlinear evaluation	Antifragile Operators: optionality, nonlinear evaluation, hybrid leadership	Antifragile Operators: optionality, nonlinear evaluation, hybrid leadership, starting small
Institutional Investment Indices: follow-up and support, champions	Institutional Investment Indices: follow-up and support, funding, policy alignment, champions	Institutional Investment Indices: follow-up and support, funding, policy alignment

## Data Availability

Not applicable.

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
