# Peer review of "Health Service Implementation and Antifragile Characteristics in Rural Communities: A Dirt Research Approach"

_ijerph, 2023, doi:10.3390/ijerph20146418_

Round 1

Reviewer 1 Report

The authors present a poorly-written, ill-defined study which lacks methodologic details and raises considerable ethical concerns.

Starting with ethical concerns:  The authors do not indicate whether the reported research what reviewed and approved by a research ethics committee.  They state that, “Informed consent was not obtained due to the informal nature of the discussion with rural residents.”  Such a statement and course of action needs to be justified and reviewed by a committee outside of the research team.  On line 190, they refer to an ethical plan, but they do not define this plan.  They indicate that they do not use direct quotes from study participants/community residents, but then go on to identify individuals (café owner, school head mistress) who would be easily identified from context by other residents and attribute statements to them.  They use the actual names of the communities they include in the study rather than pseudonyms – this minimally could produce the identity of participants.

It is not clear how “Dirt Research” differs from traditional ethnography.  The authors describe their approach as, “Central to dirt research is its experiential element where resident scholars eat, travel, play, work, and live in the communities in which they conduct their research. Through these entirely human experiences, scholars come to understand rural challenges through a new perspective - not as a researcher analyzing census data in an urban university library, but as a rural resident subject to the realities of rural living [35].”  This sounds exactly like traditional ethnography.

While the authors note the experiential element of their approach, they do not provide sufficient detail about their  Methods to allow the reader to evaluate whether they invested sufficient time any of the three communities to gain such experience.  They do not indicate the amount of time they invested in each community – other than noting that fieldwork for the Iceland and Australian communities was conducted in 2019, and for the Canadian community was conducted in 2020.  They do not indicate the number of times they visited each community, how many people they actually spoke with, how many participants completed the semi-structured interviews they note.

The authors give no idea about how they recorded information, managed their data, or analyzed their data.

The vignettes are presented as Results, but actually do not address any of the elements of the rural health delivery model they present (antifragile operators and institutional investment).  The authors mix a presentation of methods and discussion into their results (the vignettes). 

While extensive, it is not clear how the Discussion is based on the results.  Rather, the Discussion appears to be a separate presentation of their model based on study materials not included in the Results.

The writing is messy, with missing words, concepts defined and redefined, and the use of contractions.  It is not clear why so many of the sentences are in italic.  The authors misspell the person who developed the Dirt Research concept – Innes on line 174, Innis on line 175.

Perhaps the authors attempted combine a literary approach with scientific content, but the presentation fails at both.

Reviewer 2 Report

Minor issues:

1. Line 64: please correct the spelling mistake "acerbated".

2. Table 1: the authors provide "antifragile operators" with their definitions. However, it is not clear how the authors determined the operators. Are the five operators derived from a literature search? Provide the process and motivation for identifying and choosing these five operators for this manuscript.

3. Table 2: the authors identified five institutional investment indices needed for rural implementation. However, the authors do not indicate the process they followed to identify these five indices. Please provide details and/or references. 

Reviewer 3 Report

This article focuses on program implementation from communities in three countries with universal health mandates: Australia, Canada, and Iceland. This article conducted three ethnographic studies over a two-year period to understand the existence and impact of anti-fragile operators and institutional investment indices in rural communities. The results of these ethnographies provide evidence for a foundation on which a place-based rural health system can be constructed, addressing the access gap between rural and urban communities. The suggested changes are as follows:

1.     Table 2 is across pages. You need to rearrange Table 2 on one page。

2.     The article contains grammatical errors. For example,“This confidence and comfort is hard to overstate.[667] Please check the whole article carefully.

3.     This article focuses on program implementation from communities in three countries with universal health mandates: Australia, Canada, and Iceland. Therefore, the reviewer suggests that the results can be presented in table form in combination with Table 1 and Table 2 to make a concise and clear comparative analysis of the survey results of Australia, Canada and Iceland in the section of 3.Results.

Reviewer 4 Report

A very well written manuscript with an interesting focus which I think is relevant for a large public. I just have a few questions/comments.

-Do the authors belive it is possible to replicate this study?

-How have the authors worked with ethical guidelines regarding information to the participants (informants who were the basis for the vignettes) especially in small communities where people such as the principal can be identified.

-I am not really clear of the meaning in line 648 "The vignettes developed through dirt research and presented here aim to prepare the reader for undertaking an implementation of an eHealth initiative in a rural community. That undertaking should make the reader feel appropriately uncomfortable, as the complexity of these implementations is clear. " My question is why should this undertaking should make the reader feel uncomfortable and why is this approproately?

-Line 630 "Representation in rural health research is important and contributes to the growing need for place-based health system reform. Place-based reform could prioritize equitable care where every individual has equal opportunity to be healthy [45]. The reference 45 is from the Australian Health Care system and the conclusion for this statement might be to strong..see my suggestion  in red 

Good luck!

Round 2

Reviewer 1 Report

The authors have vastly improved the presentation of their manuscript.

Ethical concerns remain.  The authors argue that they could not get ethics committee review for their research due to its informal nature.  They state, “The informal nature of the work did not satisfy the rigour of a formal ethics protocol, although clearance for the overall project components was obtained and is now noted in the document.”  First, the logic of this approach is suspect.  Ethnographic research, often as informal as that described by the authors, continues to be conducted and must satisfy the rigors of a formal ethics protocol.  Ethics boards approve the approach; a formal ethic protocol does not require that every conversation be approved.  Ethics boards can wave the requirement of formal informed consent.  Second, the authors note that their “study was approved by the Carleton University Research Ethics Board (Protocols #108875 on September 26, 2020 #110145 on April 15, 2021).”  However, they also report that their research in Thorfjörður, Iceland was conducted in 2019, and that field visits to Lemur and Ericstown were conducted in 2019.  I do not understand how ethics approval can be obtained one or two years after the data were collected.  These issues must be addresses before the revision is reviewed further.

The justification for calling their approach “dirt research” remains suspect.  They continue to call their approach “2.1. Ethnographic Methodology.”  Rapid ethnographic assessment has a long-history.
